# Current and Future Drug and Device Therapies for Pediatric Heart Failure Patients: Potential Lessons from Adult Trials

**DOI:** 10.3390/children8050322

**Published:** 2021-04-22

**Authors:** Bibhuti B. Das, William B. Moskowitz, Javed Butler

**Affiliations:** 1Heart Center, Department of Pediatrics, Mississippi Children’s Hospital, University of Mississippi Medical Center, Jackson, MS 39212, USA; wmosckowitz@umc.edu; 2Department of Medicine, University of Mississippi Medical Center, Jackson, MS 39212, USA; jbutler4@umc.edu

**Keywords:** acute heart failure syndrome, pediatric heart failure, pharmacotherapy for heart failure, device therapy for chronic heart failure

## Abstract

This review discusses the potential drug and device therapies for pediatric heart failure (HF) due to reduced systolic function. It is important to realize that most drugs that are used in pediatric HF are extrapolated from adult cardiology practices or consensus guidelines based on expert opinion rather than on evidence from controlled clinical trials. It is difficult to conclude whether the drugs that are well established in adult HF trials are also beneficial for children because of tremendous heterogeneity in the mechanism of HF in children and variations in the pharmacokinetics and pharmacodynamics of drugs from birth to adolescence. The lessons learned from adult trials can guide pediatric cardiologists to design clinical trials of the newer drugs that are in the pipeline to study their efficacy and safety in children with HF. This paper*’*s focus is that the reader should specifically think through the pathophysiological mechanism of HF and the mode of action of drugs for the selection of appropriate pharmacotherapy. We review the drug and device trials in adults with HF to highlight the knowledge gap that exists in the pediatric HF population.

## 1. Introduction

A working definition of heart failure (HF) in children is “a progressive clinical and pathophysiological syndrome caused by cardiovascular and noncardiovascular abnormalities that results in characteristic signs and symptoms including edema, respiratory distress, growth failure, and exercise intolerance and accompanied by circulatory, neurohormonal, and molecular derangements” [1]. In adults, HF can occur with preserved ejection fraction (HFpEF) or with reduced ejection fraction (HFrEF). This review addresses only HF due to reduced systolic function, which is conventionally reported by left ventricular (LV) ejection fraction in percentage. Current pharmacological therapies for HF in children is extrapolated from adult cardiology practices rather than evidence from controlled clinical trials. However, there are significant barriers to applying adult data to children because of tremendous heterogeneity in the mechanism of HF and variations in the pharmacokinetics and pharmacodynamics of drugs from birth to adolescence. Simultaneously, there are significant challenges in performing well-designed drug trials in children with HF because of difficulty achieving sufficient enrolment and heterogeneity in HF causes.

This review discusses the current and future pharmacological therapies in children with acute and chronic HF, the mechanism of action of drugs, and the need for future clinical trials in children for the safety and efficacy of newer drugs that are used in adults. Furthermore, we discuss the device therapies in adult HF and highlight the potential of these devices for pediatric HF, as we learn from adult trials.

## 2. Acute Heart Failure Syndrome

Acute HF syndrome (AHFS) is described as a structural or functional alteration in the heart that occurs rapidly, followed by congestion, malperfusion, hypotension, and end-organ dysfunction resulting in a need for hospitalization and urgent therapy [2]. The goals of acute HF management in children are to improve hemodynamics and prevent progression (Figure 1). Current management includes stabilization with intravenous inotropes/vasopressors, diuretics, mechanical ventilation, treatment of arrhythmia, progression to mechanical circulatory support, and heart transplantation if needed [3].

### 2.1. Diuretics

The management of AHFS relies on an accurate assessment of the patient’s congestion and adequacy of systemic perfusion [4]. Adequate diuresis is most commonly achieved by loop diuretics (furosemide and bumetanide) intravenously as the first therapy line. They act by inhibiting the sodium-potassium-chloride co-transporter on the ascending limb of the loop of Henle. This results in decreased reabsorption of sodium, potassium, chloride, and water. In cases where loop diuretics are not adequate alone, thiazide (chlorothiazide and metolazone) diuretics (which inhibit the sodium-chloride co-transporter in the distal convoluted tubule and act synergistically with loop diuretics to amplify sodium and water loss) are recommended as per the consensus statements for the treatment of pediatric HF by the International Society for Heart and Lung Transplantation (ISHLT) [5]. One of the neurohumoral responses in AHFS is excess vasopressin release from the hypothalamus, which may cause hyponatremia. In this circumstance, vasopressin receptor (V_2_) antagonists (tolvaptan and conivaptan) can be used to enhance free-water excretion and correct hyponatremia. The EVEREST study in adults with HF demonstrated that tolvaptan improves edema, body weight, dyspnea, and sodium level, but there are no survival benefits [6]. Tolvaptan has been shown in small series to increase urine output and improve serum sodium concentration in children with HF [7,8].

### 2.2. Vasoactive Drugs

Vasoactive drugs are used as a rescue therapy in AHFS to improve systemic perfusion and prevent end-organ dysfunction. These drugs improve myocardial contractility and when combined with appropriate blood pressure control, they may increase cardiac output. However, the use of vasoactive drugs in children with AHFS is mainly used as a bridge to transplant or mechanical circulatory support and is not recommended for maintenance therapy or recovery of heart muscle function [5]. The conventional vasoactive drugs used are dopamine, milrinone, and epinephrine. Dopamine and epinephrine are sympathomimetic agents and target myocardial β-adrenergic receptors. The cyclic adenosine monophosphate (cAMP) pathway is activated, enhancing calcium release from the sarcoplasmic reticulum, which binds to troponin-C augmenting myocardial contractility enhancing actin-myosin interaction. However, the drawback is that β1-stimulation also increases heart rate and myocardial oxygen consumption. Milrinone is a phosphodiesterase-3 inhibitor (PDEI) and is the most commonly used vasoactive agent in children to increase contractility and afterload reduction through vasodilation. Because calcium reuptake is also cAMP-dependent, PDEI also enhances diastolic myocardial relaxation, resulting in decreased filling pressure. Because milrinone can cause hypotension, a combination of low dose epinephrine or dopamine and milrinone can be used to balance blood pressure and cardiac contractility [9].

### 2.3. Levosimendan

Levosimendan can be an alternative to milrinone for AHFS. It is not available in the US but is widely used in Europe. It is a calcium-sensitizing agent that binds to troponin-C, enhancing its sensitivity to intracellular calcium, and has positive inotropic action. It also opens up the adenosine triphosphate (ATP)-dependent potassium channels leading to smooth muscle relaxation, vasodilation, and decreased systemic vascular resistance. The hemodynamic effects of levosimendan include increased cardiac output and decreased filling pressure. It causes an increase in contractility without an increase in myocardial oxygen demand and has lusitropic action on the myocardium. Initial studies of levosimendan in adults (LIDO [10], RUSSLAN [11], and CASINO [12]) have suggested that this drug might improve the prognosis of patients with AHFS. Two subsequent large trials (SURVIVE [13] and REVIVE [14]) showed that levosimendan improves the symptoms of HF but does not improve survival. The data in adults led to the trial of levosimendan in children. Pediatric patients who received levosimendan can be divided into two groups: the first group who received levosimendan as prophylaxis for low cardiac output in the post-operative period [15,16] for whom there was no significant benefit of the drug; and the second group with end-stage HF and inotrope dependency who received this drug and showed improved status in terms of inotropes requirement and hospital length of stay [17,18]. The duration of levosimendan therapy is only 24 h. Levosimendan’ s role in children is still unconfirmed, but it appears to be effective in AHFS but play no role as a prophylaxis to prevent HF.

### 2.4. Nesiritide

Nesiritide, a recombinant human B-type natriuretic peptide (BNP), is identical to the endogenously produced BNP. It increases intracellular cyclic GMP (cGMP), resulting in vasodilation and a decrease in afterload. It inhibits neurohormonal activation, produces diuresis, natriuresis, and increases cardiac output [19]. The efficacy and safety of nesiritide in acute HF management have been demonstrated in adults [20]. There are small studies in children that have shown the usefulness of nesiritide in AHFS [21,22,23]. The ISHLT pediatric HF consensus guidelines have recommended that nesiritide be considered as adjunctive therapy in critically ill children with biventricular dysfunction suffering from edema and oliguria despite standard HF management [5]. Nesiritide has been discontinued in the US market since 2018 after the ASCEND-HF trial showed no impact on recovery or HF hospitalizations, 74% increased risk of 30-day mortality, and 54% increased risk of worsening renal function [24].

## 3. Chronic Heart Failure Syndrome

After presenting with AHFS, when the hemodynamics stabilize and end-organ functions recover, the child is often left with the lingering diagnosis of chronic HF syndrome (CHFS), with or without a genetically based or syndrome/systemic disease-based diagnosis. Several regulatory neurohumoral and counter-regulatory pathways are involved in the pathogenesis of CHFS (Figure 2).

The goal of the clinical management of CHFS in children is to permit recovery, maintain stability, prevent progression, and provide a reasonable milieu to allow somatic growth and optimal development into adult life. Typically, a multi-drug approach is required, with either sequential combination therapy or upfront combination of angiotensin-converting enzyme inhibitors (ACEi), β-blockers, diuretics, and aldosterone antagonists. Herein we review the current and future landscape of drug therapies for chronic HF in adults with evidence for randomized clinical trials to highlight the knowledge gap in pediatric HF. Contrary to adults, the clinical trials to verify the safety and efficacy of drugs used in CHFS in children are limited (Table 1).

### 3.1. ACE Inhibitors and Aldosterone Antagonists

Renin-angiotensin-aldosterone system (RAAS) pathway activation is a central physiologic response to decreased renal perfusion and is characteristic of CHFS with reduced systolic function. Renin stimulates the conversion of hepatic angiotensinogen to angiotensin I, which is acted upon by the angiotensin-converting enzyme (ACE), predominantly in the lungs, to form angiotensin II, a potent vasoconstrictor that causes chronic deleterious effects in HF. These include the vasoconstriction of renal afferent arterioles and the release of aldosterone, which causes an increase in sodium reabsorption. When the RAAS activation is undeterred, vasoconstriction (increased afterload) leads to cardiomyocyte hypertrophy and apoptosis [28]. Furthermore, RAAS has a significant profibrotic effect on cardiac tissue by stimulating metalloproteinase and promoting endothelial dysfunction.

Clinical trials of adult patients with HF with reduced systolic function (CONSENSUS in 1987 [29] and SOLVD-treatment trial in 1991 [30]) demonstrated that enalapril improved survival, stabilized LV size, and reduced the progression of HF symptoms and hospitalizations in both asymptomatic and symptomatic patients with HF and reduced systolic function. The most extensive prospective study in children with single ventricle physiology with the right ventricle (RV) as systemic ventricle and preserved ejection fraction was conducted by Hsu et al. [26]. This study did not prove a clinical benefit of ACEi treatment; however, these data cannot be transferred to children with cardiomyopathy and other etiologies like cardiomyopathies with reduced systolic function. A retrospective study in 2010 detected a positive benefit of ACEi in children with dilated cardiomyopathy compared to no treatment; however, treatment with digoxin/diuretic revealed comparable results [31]. Despite the lack of prospective, randomized, blinded, placebo-controlled trials in children with dilated cardiomyopathy, treating children with CHFS with ACEi is widely accepted and recommended by ISHLT to manage HF [5]. Pitt and colleagues demonstrated the independent efficacy of spironolactone as an additive to patients’ survival benefit on ACEi therapy in the RALES trial [32]. The use of an aldosterone antagonist such as spironolactone or eplerenone is also recommended in addition to ACEi (level of evidence B) in children [5]. The risk of hyperkalemia in patients treated with both ACEi and aldosterone antagonists can be safely managed with anticipatory monitoring. In boys with dilated cardiomyopathy due to Duchenne muscular dystrophy, eplerenone reduced LV dilation progression and dysfunction compared to the placebo [33].

### 3.2. Angiotensin Receptor Blockers

The effect of angiotensin receptor blockers (ARBs) in HF is similar to that of ACEi. For patients intolerant of ACEi (due to idiosyncratic side effects, angioedema, or bradykinin-system-related cough), an angiotensin receptor blocker (ARB) such as valsartan, losartan, or candesartan is often well-tolerated and effective [34]. Because ARBs do not block the degradation of kinins, angioedema and persistent cough are uncommon. Nevertheless, ARB has not replaced ACEi as it has not proven superior in preventing adverse events in adults [35]. Because ACEi incompletely inhibits angiotensin, combinations of ARB and ACEi have also been evaluated. In the Valsartan Heart Failure Trial, a combination of valsartan and ACEi decreased hospitalization risk in adults but increased mortality when such a combination was used in conjunction with beta-blockers [36]. Because combinations of ACEi and ARB have a higher likelihood of renal side effects, such combinations are not used in children.

### 3.3. β-Adrenergic Receptor Blockers

In response to HF, the body compensates by activating the sympathetic nervous system, increasing circulating catecholamines [28]. The physiologic effects of catecholamines are mediated by the activation of α- and β-adrenergic receptors. The activation of β1receptors, which are found primarily within the myocardium, results in acutely increased heart rate, cardiac contractility, and more rapid atrioventricular conduction. The activation of β2 receptors, which are present in the heart but more prominently located in bronchial and peripheral vascular smooth muscle, results in vasodilation and bronchodilation. However, chronically increased sympathetic activation is detrimental to the failing heart, resulting in excessive myocardial oxygen consumption, leading to myocardial fibrosis and apoptosis. The β-adrenergic receptor blockade aims to antagonize the deleterious effects of chronic sympathetic myocardial activation and reverse LV remodeling.

Numerous randomized controlled trials in adults have validated the beneficial effects of the β-adrenergic blockade in reducing symptoms, hospitalizations, and mortality related to chronic HF [37,38,39]. β-adrenergic blockers that are approved for adult use are bisoprolol, metoprolol, and carvedilol. The first and only randomized, controlled study in pediatric HF that addressed carvedilol for HF management [25] failed to show the benefit of the treatment, unlike the adult studies. Despite the negative outcomes, that study highlighted several issues faced in pediatric cardiac research, including the underpowered study, the difficulties in interpreting results from a heterogeneous population, and the importance of designing an appropriate endpoint. Several retrospective studies showed that carvedilol improved the ejection fraction and improved the children*’*s HF clinical status [40,41,42]. The ISHLT guidelines recommend that it is “reasonable to consider initiation of β-blockade in children with LV systolic dysfunction” [5].

### 3.4. Angiotensin Receptor-Neprilysin Inhibitor

At the cellular level, the compensatory gain in cardiac excitation-contraction coupling mediated by sympathetic stimulation ultimately becomes unsuccessful, as the sustained leak of calcium from the sarcoplasmic reticulum leads to the depletion of intracellular calcium and ultimately impairs contractility [43]. Several peptides such as natriuretic peptides, bradykinin, or adrenomedullin decreased the harmful effects of RAAS and activated the sympathetic nervous system. Natriuretic peptides are coupled to and activate guanyl cyclase A, which increases the intracellular concentrations of the second messenger, cGMP. The latter, in turn, activates protein kinase C, leading to vasorelaxation, natriuresis, and diuresis. Atrial natriuretic peptide and BNP also inhibit renin secretion and aldosterone production and attenuate cardiac and vascular remodeling, apoptosis, ventricular hypertrophy, and fibrosis [44]. Typically, these compensatory actions are insufficient to prevent or stop HF development because the enzyme neprilysin readily destroys natriuretic peptides. Neprilysin levels are increased in chronic HF, and, thus, the clearance of these neuropeptides is accelerated [45]. Inhibition of neprilysin enhances the effects of endogenous natriuretic peptides, which exerts vasodilation, anti-hypertrophic/anti-fibrotic action, mitigating the detrimental effects of angiotensin, endothelin, and aldosterone. Concomitant inhibition of angiotensin synthesis action is significant because neprilysin inhibition alone is accompanied by activation of the RAAS, possibly because angiotensin itself may be a substrate for neprilysin [46]. Although ACEi may attenuate angiotensin’s actions, simultaneous ACE and neprilysin blockade can lead to severe angioedema. This has been shown by the Omapatrilat versus Enalapril Randomized Trial of Utility in Reducing Events trial [47]. The preferred approach to modulate the neuroendocrine system’s balance is by combining a neprilysin inhibitor while simultaneously blocking the effects of angiotensin action by an ARB. The combination of valsartan (an ARB) and sacubitril (neprilysin inhibitor) is more effective than the ACEi or ARB alone because of the incremental benefits of neprilysin inhibition in HF in reducing the risks of both sudden cardiac death and death from worsening HF [48]. Neprilysin inhibition may have additional benefits, including improved hemodynamics, reduced ventricular wall stress, myocardial fibrosis, ventricular hypertrophy, and attenuation of progressive ventricular remodeling [49]. Further, neprilysin inhibition of sympathetic drive or potentiation of vagotropic effects in HF is well documented, although it is just coming to prominence as an effective therapy.

Based on the PARADIGM-HF trial’s success, the pediatric HF community has conducted a PANORAMA-HF study trial comparing sacubitril/valsartan to enalapril in children with Class C HF whose systemic ventricle is a LV with an ejection fraction <40% [50]. This study has now completed enrolment in three age strata (6 years to 18 years; 1 year to 6 years; and 1 month to 1 year). Surprisingly, during the ongoing enrolment in PANORAMA-HF, in October 2019, the FDA approved the use of sacubitril/valsartan for use in children older than 1 year of age with symptomatic HF with reduced systolic function. The explanation given was that “the approval was based on an analysis at 12 weeks from the 52-week PANORAMA-HF trial which demonstrated reductions in the cardiac biomarker N-terminal proBNP (NT-proBNP) in pediatric patients 1 to <18 years with HF due to systemic LV systolic dysfunction with sacubitril/valsartan. Because sacubitril/valsartan improved outcomes and reduced NT-proBNP in adult patients in PARADIGM-HF, this effect on NT-proBNP was considered a reasonable basis to infer improved cardiovascular outcomes in pediatric patients. The reductions from baseline in NT-proBNP for sacubitril/valsartan (44%), and the active comparator enalapril (33%), were similar to or greater than those observed in adults, but importantly, the difference between treatment groups was not statistically significant. Safety and tolerability of sacubitril/valsartan in pediatric patients were consistent with that observed in adult patients.” The effect of inhibition of neprilysin on the level of NT-proBNP is minimal as it is cleared from the circulation mainly by neuropeptide receptors [51], making this a valuable biomarker to measure and track, as a decrease in NT-proBNP may then be attributable to improved HF and not a drug effect.

For adults, valsartan/sacubitril is available in three dosage strengths: 24/26 mg, 49/51 mg, and 97/103 mg. These doses are 50 mg, 100 mg, and 200 mg in the clinical trial [48]. The target maintenance dose of valsartan/sacubitril is 97/103 mg twice daily as tolerated in adult patients. In pediatric patients, the target maintenance dose is dependent on body weight. In children <40 kg, the starting dose should be 1.6 mg/kg of the combined amount of both valsartan and sacubitril. The dose is titrated every two weeks upward from 2.3 mg/kg up to a max dose of 3.1 mg/kg based on tolerance. An oral solution can also be compounded for use in children. In most HF drugs, the dose is based upon body weight in the pediatric population. (Table 2).

### 3.5. Ivabradine

Ivabradine, an I_f_ current inhibitor in the sinoatrial node, acts by reducing heart rate and helped reduce HF hospitalization and death from HF in adults in the SHIFT and BEAUTIFUL trials [52]. The safety of ivabradine has been validated in a pediatric phase II/III dose-finding clinical trial of children with stable HF [27]. In this study, ivabradine resulted in a reduction in heart rate, an increase in systolic function, and a trend towards improved quality of life. There was no significant difference in NT-pro BNP levels between ivabradine and placebo in this study. Based on adult studies, further control of heart rate in children with HF with or without β-blocker may improve the outcome.

### 3.6. Omecamtiv Mecarbil

Omecamtiv mecarbil is a new class of myotropes that binds selectively to the cardiac myosin protein base and permits ADP-P release from the myosin-actin-ATP complex, thus increasing the number of myosin heads that can bind to the actin filament and facilitate cardiac sarcomere contractility. As a result, there is increased systole duration but no increase in myocyte calcium, and no increase in myocardial oxygen consumption, unlike the vasoactive agents [53]. The GALACTIC-HF trial has shown a lower incidence of a composite HF event or death from cardiovascular causes than those who received a placebo in adults with HF with reduced systolic function [54]. At present, there are no data available regarding the efficacy of this drug in children, but as in other drugs, omecamtiv mecarbil may be an effective alternative to improve outcomes in pediatric systolic HF in the future.

### 3.7. Sodium-Glucose co-Transporter 2 (SGLT-2) Inhibitors

The DAPA-HF trial [55] and the EMPEROR-Reduced trial [56] have shown that SGLT-2 inhibitors (dapagliflozin and empagliflozin) reduced the risk of worsening HF events in adults with reduced EF, irrespective of the presence of diabetes at baseline. The precise mechanism of SGLT-2 inhibition in achieving its effect remains uncertain, although a modest reduction in central venous pressure has been demonstrated [57,58]. No data are available in children, but this class of drugs appears to be helpful in adult clinical trials. This drug may be helpful in the future, especially in children and young adults with HF who are also obese and have metabolic syndrome.

### 3.8. Vericiguat

The cGMP pathway has been implicated as an essential regulator of endothelial function in both primary and secondary pulmonary hypertension and is relevant to myocardial and vascular smooth muscle dysfunction in HF states as well [59]. Vericiguat is a cGMP pathway stimulator, acting directly on intracellular soluble guanylyl cyclase to increase cGMP production, independent of endogenous nitric oxide production (typically depressed in HF). In the VICTORIA trial, adult patients with chronic symptomatic HF and an ejection fraction of <45% had a reduced risk of cardiovascular death, all-cause death, and HF hospitalization [60]. No data are currently available for children, but it seems promising, especially in bi-ventricular HF and HF associated with congenital heart disease.

### 3.9. Digoxin

Digoxin is a cardiac glycoside and was the cornerstone of HF therapy for decades until a paradigm shift in HF pathophysiology led to a shift from inotropic therapy to neurohormonal modulation. Digoxin binds to the sarcolemmal Na^+^-K^+^ ATPase pump, thereby preventing Na^+^ removal from the myocytes in the exchange of K^+^. As a result, more intracellular Na^+^ available for calcium influx through the Na^+^-Ca^++^ exchanger. Calcium is transported into the sarcoplasmic reticulum and helps a more muscular mechanical contraction. Furthermore, evidence from experimental studies suggests that digoxin binds directly to ryanodine receptor-2 [61]. Digoxin also exhibits negative chronotropic action by increasing cardiac vagal tone and improves symptomatic HF in children. Recently, the use of digoxin in HF has declined because of potential adverse effects such as polymorphic ventricular tachycardia despite earlier randomized clinical trial data showing significant improvement in ejection fraction and exercise capacity in studies such as PROVED [62], RADIANCE [63], and DIMIT [64]. After digoxin’s success in earlier trials that included only a small number of patients, the Digitalis Investigation Group (DIG) randomized, double-blind, placebo-controlled trial, including 5800 patients, concluded that there was no difference in all-cause mortality with the addition of digoxin to ACEi and diuretics [65]. Because of the uncertain benefits of digoxin and severe pro-arrhythmic action, the use of digoxin declined recently both in adults and children with HF. Subsequently, in a retrospective analysis of the Valsartan in HF Trial (Val-HeFT), digoxin treatment was associated with a higher risk of all-cause mortality and HF-related hospitalization. Recently, another adult HF trial studied the effect of digoxin in all-cause mortality and HF hospitalization in 2891 patients with newly diagnosed systolic HF and reported adding digoxin to ACEi β-blockers had a higher incidence of death in the digoxin group [66].

The use of digoxin in pediatric HF patients is largely empirically based but, in general, has been replaced by newer HF therapies for better neurohormonal modulation. However, a Pediatric Heart Network study recently showed that digoxin decreased interstage mortality significantly in infants with single ventricle physiology [67]. Digoxin is also helpful in HF with atrial arrhythmia and the control of heart rate. Therefore, in our view, digoxin should not be discarded from the HF armamentarium. Digoxin probably still plays a role in patients with severe HF who cannot tolerate ACEi or β-blockers due to lower blood pressure/renal dysfunction. Digoxin has a role specifically to reduce symptoms and hospitalization in children with CHD and HF. It is essential to closely monitor the creatinine and potassium levels to minimize the risk of digoxin toxicity.

## 4. Clinical Trials: What’s on the Horizon?

A few other drugs are under trial for AHFS in adults, including istaroxime (Horizon-HF trial [68], serelaxin (RELAX-AHF-2 trial) [69], and ularitide (TRUE-AHF trial) [70], and outcome results are pending. There is no experience of these drugs in pediatric HF. In the future, there are opportunities to explore the role of renin inhibitors, calcineurin, and the role of calcium modulation in the myocytes and their roles in myocardial function and HF remain to be explored. Although our understanding of HF’s etiology and pathophysiology in children has improved, the efficacy and safety of newer pharmacological therapies remain uncertain.

## 5. Innovative Therapies for Acute and Chronic HF

Chronic HF is a progressive disease, meaning it continues and worsens. Despite the recent success of drug therapy and the PANORAMA-trial in children to access valsartan/sacubitril for chronic HF with reduced systolic function in children, there remains residual risks of mitral regurgitation, fluid overload due to resistance to standard diuretics, autonomic dysregulation, etc. In adults, the most innovative device therapies are driven by industry as the FDA has expedited access to innovative devices for the diagnosis and treatment of serious illnesses, such as HF. The Centers for Medicare and Medicaid Services increased hospital reimbursement for these technologies to increase access to breakthrough technologies [71]. A judicious balance between extrapolation from adult device therapy guidelines and child-specific data development will be a wise approach to optimize pediatric HF management. This approach has been helpful as reflected by the increasing role of ventricular assist devices (VAD) in managing end-stage HF in children.

### 5.1. Left Ventricular Assist Devices

Mechanical circulatory support with left ventricular assist devices (LVADs) are increasingly used to manage HF in infants and children as a bridge to transplantation or destination therapy in selected patient cohorts [28]. In children, Berlin Heart EXCOR**^®^** (Berlin Heart GmbH, Berlin, Germany) and HVAD**^®^** (HeartWare Inc., Framingham, MA, USA) are used with increased survival benefits and successfully bridged to heart transplantation [72,73]. Recently, the FDA approved HeartMate 3**^®^** (Abbott, IL, USA) for children >19 kg based on the consortium data [74]. Although VAD support and durability in pediatric patients are getting better, a high rate of serious adverse events persists, including infection, bleeding, sensitization, device malfunction, and neurologic injury in children [75]. There are upcoming devices that are fully intracorporeal without an external driveline and may alleviate some of these complications in the future.

### 5.2. Regenerative Strategies

The potential for cardiac regeneration in children may be greater than in adults. It has been shown that there is myocyte regeneration after unloading of the dilated LV by mechanical circulatory support in pediatric dilated cardiomyopathy patients with end-stage HF [76]. The regenerative strategy is the basis of reversible pulmonary artery band (PAB) in infants and young children with dilated cardiomyopathy [77,78]. Application of a reversible PAB for 2–3 months can increase the contractility of LV by ventricle–ventricle interaction as both RV and LV share a common septum. However, a randomized, prospective, clinical trial is warranted to study the role of reversible PAB in pediatric DCM patients to diminish the need for mechanical circulatory support and cardiac transplantation.

There has been a rapid proliferation of clinical studies using stem cells in adults with HF [79], yet little convincing evidence of clinically significant improvement. Unlike adult trials, there are no controlled studies in children with HF. Future clinical trials in children should be based on lessons learned from adult trials, as both efficacy and safety are needed.

### 5.3. Devices to Treat Mitral Regurgitation

The prognosis in HF patients who develop secondary mitral regurgitation (MR) due to dilatation of LV leading to annular dilatation of mitral valve (MV) is worse, with a direct quantitative relationship between the severity of MR and both death and hospitalizations for HF in adults [80]. The MitralClip**^®^** (transcatheter MV Repair) (Abbott, IL, USA) device was approved by the FDA in 2013 for adult patients who cannot tolerate surgery of the MV. The Cardiovascular Outcomes Assessment of the MitralClip**^®^** in Patients with HF and Secondary MR (COAPT) trial has provided a new option for managing selected adult patients with secondary MR and HF [81]. Many other new trials are ongoing, including APOLLO [82] and SUMMIT [83], to investigate the safety and efficacy of transcatheter MV replacement. No MV device has been used in children, and in the future, adolescents and young adults may qualify for transcatheter therapy of the MV.

### 5.4. Modulation of Autonomic Nervous System

Autonomic dysregulation (i.e., parasympathetic system withdrawal and sympathetic system overactivation) leading to persistent tachycardia, increased oxygen demand, and higher afterload can cause the development and progression of HF [84]. Afferent input to the baroreflex originates from the carotid sinus and aortic arch receptors, which are stimulated by arterial distension. The baroreflex regulates the efferent sympathetic and parasympathetic output via the rostral ventrolateral medulla and nucleus ambiguous [85]. Baroreflex activation therapy stimulates the carotid baroreceptor and optimizes baroreflex dysfunction, which can subsequently control autonomic dysregulation. Based on adult studies [86], the FDA approved the Barostim**^®^** neo system (CVRx, Inc., Minneapolis, MN, USA) in 2019 to improve HF symptoms in adults. There is no experience of baroreflex activation therapy in pediatric HF.

### 5.5. Electrophysiological Modulation of Cardiac Contractility

Chronic HF due to decreased systolic function is characterized by abnormal intracellular calcium handling and mechanical and electrophysiological dysfunction [87]. The FIX-HF-5C (Evaluate Safety and Efficacy of the OPTIMIZER**^®^** System in Subjects with Moderate-to-Severe Heart Failure) trial has shown promising results in the symptomatic improvement of chronic HF in adults [88]. The cardiac contractility modulation therapy (OPTIMIZER**^®^** System) delivers a biphasic, long-duration (~20 milliseconds), high-voltage (~7.5 V) electrical signal to the septum of the RV during the absolute refractory period [89,90]. As the delivery of the electric signals occurs during the refractory period, it does not result in myocardial contraction but leads to structural and functional changes in the myocardium, including molecular changes, enhanced contractility, and decreased LV volume [91]. There is no experience of cardiac contractility modulation therapy in pediatric HF.

### 5.6. Inter-atrial Devices to Create Shunt

The elevated left atrial pressure is the sine qua non of chronic HF, which is directly related to exercise intolerance. In adults, various minimally invasive devices such as the Inter Atrial Shunt Device (IASD**^®^**, Corvia Medical) [92] and the V-Wave**^®^** (Casarea, Israel) interatrial shunt device [93] have been used to create a permanent controlled left-to-right shunting to decompress the left atrium. There are ongoing randomized controlled trials using these devices to study the efficacy and safety in adult HF patients. These devices have not been used in children, although atrial septostomy and creating an inter-atrial shunt is a common practice to decompress the left atrium in children with acute HF while supported with ECMO.

### 5.7. Peritoneal Direct Sodium Removal for Volume Management

Chronic HF patients develop resistance to diuretics, and in adults, peritoneal direct sodium removal is a novel approach to treat volume overload. The RED DESERT study is an ongoing trial to investigate the feasibility and safety of the Alfapump**^®^** (Sequana Medical NV, Ghent, Belgium) peritoneal direct sodium removal system in adults with chronic HF [94]. There is no experience of Alfapump**^®^** in pediatric HF.

## 6. Conclusions

The ACC/AHA/HFSA guidelines for pharmacological therapy for HF in adults were updated in 2016 [95], and the new adult HF medications are being used in children without clear evidence of safety and efficacy. This is likely to change after the success of the PANORAMA-HF trial. The endpoint in this trial may be the critical element in pediatric HF research success in the future, as the use of mortality or all-cause events as an endpoint is unlikely to be an achievable goal given the number of patients required. It has been suggested that future randomized studies in children should probably include cause-specific outcomes such as a decrease in pro-BNP or BNP in children following the adult PARADIGM-HF trial model. Furthermore, pediatric HF drug trials should be global because of small numbers and etiologies’ heterogeneity. Adult HF clinical trials may serve as guidelines to design compelling drug trials in children but are not substitutes. In the meantime, it is necessary to specifically think through the mechanism of HF and the mechanism of pharmacotherapy for selecting appropriate drugs in children with HF. In children, due to small case numbers of HF, there are no incentives for the industry to develop children-specific HF therapies. The federal government should support such clinical trials to expedite the testing of drug and device therapies in pediatric HF. Novel clinical trial designs may be considered that allow for early market access by accelerating the development, assessment, and review processes, and linking reimbursement from the Centers for Medicare and Medicaid Services to FDA marketing approval. Future device-based therapies in children may minimize the side effects from pharmacotherapy and improve compliance and overall outcomes of HF in children.

## Figures and Tables

**Figure 1 children-08-00322-f001:**
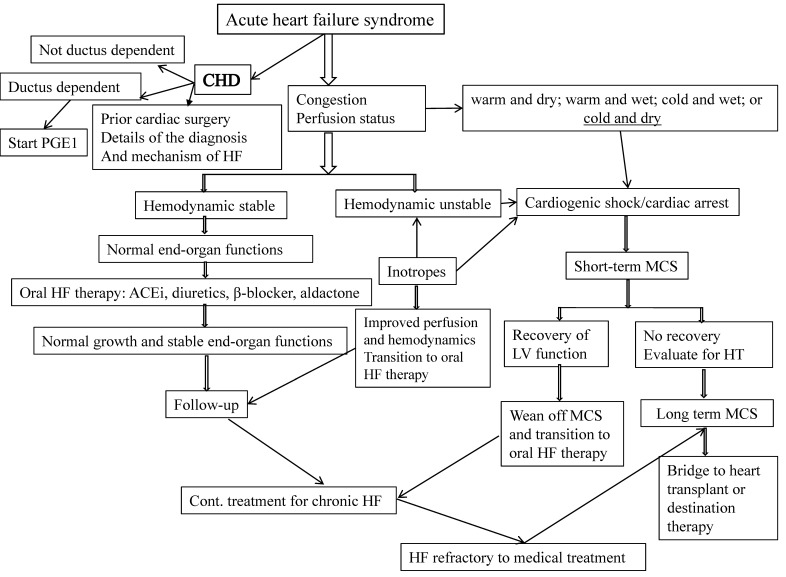
Approaches to acute HF in infants and children. (MCS: mechanical circulatory support; HF: heart failure; CHD: congenital heart disease; H/O: history of; PGE1: prostaglandin 1; ACEi: angiotensin-converting enzyme inhibitor).

**Figure 2 children-08-00322-f002:**
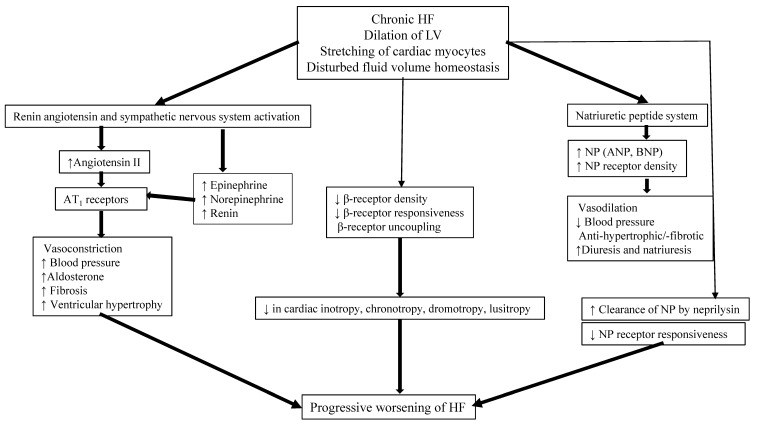
Pathophysiology of chronic HF. (LV: left ventricle; AT1: angiotensin 1; NP: natriuretic peptide; ANP: atrial natriuretic peptide; BNP: B-type natriuretic peptide).

**Table 1 children-08-00322-t001:** Limited clinical trials in children.

Title	Journal/Year (Reference)	Key Findings
Carvedilol for children and adolescents with HF. A randomized control trial	JAMA, 2007 [25]	N = 161; no significant difference between treatment vs. placebo group in the primary endpoint (clinical improvement) or secondary endpoint (ventricular function or serum BNP).
Safety of enalapril in infants with single ventricle (SV) physiology, multicenter randomized trial	Circulation, 2010 [26]	N = 230; no improvement in somatic growth, ventricular function, or heart failure severity. Routine use of enalapril not recommended in SV patients.
Ivabradine in children with DCM and symptomatic chronic HF trial: a randomized, double-blind, placebo-controlled trial with 12-months follow-up	JACC, 2017 [27]	N = 116; primary endpoint reached by 51 of 73 children taking Ivabradine (70%); Ivabradine safely reduced the resting heart rate of children with chronic HF and dilated cardiomyopathy; improvement in ejection fraction, functional class, and NT-pro BNP was noted.

**Table 2 children-08-00322-t002:** Summary of commonly used pediatric HF drugs and doses.

	Standard Pediatric Doses
**Diuretics**	
1. Furosemide	1 mg/kg dose BID up to max 6 mg/kg/day
2. Chlorothiazide	10 mg/kg dose BID up to max 2 gm/day
3. Metolazone	0.1 mg/kg dose BID up to max 20 mg/day
**Digoxin**	3 to 5 mcg/kg dose BID
**Angiotensin-converting enzyme inhibitors**	
1. Captopril	0.1 mg/kg dose TID up to max 2 mg/kg/dose
2. Enalapril	0.1 mg/kg dose BID up to max 0.5 mg/kg/day
**Beta-blockers**	
1. Metoprolol	0.1 mg/kg dose BID up to max 1 mg/kg dose
2. Carvedilol	0.025 mg/kg/dose BID up to max 0.5 mg/kg/dose BID
**Aldosterone antagonist**	
Spironolactone	1 mg/kg dose BID up to max 200 mg/day

(BID = twice daily, TID = three times daily, max = maximum, mg/kg = milligram per kilogram).

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
