# Peer review of "Current and Future Drug and Device Therapies for Pediatric Heart Failure Patients: Potential Lessons from Adult Trials"

_children, 2021, doi:10.3390/children8050322_

Round 1

Reviewer 1 Report

The Authors of this review focuses on the HF pharmacotherapy in children mainly using extrapolations from adult cardiology practices or guideline suggestions. The review is well written. However, to increase its clinical impact the Authors should address the following issues:

1) The therapy should be addressed according to HF categorization, including reduced or preserved ejection fraction phenotypes. Some specifc heart diseases subtending HF in children may deserve brief comment (e.g. obstructive hypertrophic cardiomyopathy

2)Pharmacodynamics peculiarities and recommended doses of HF medications in children vs. adults deserves comment

3) The treatment of arrhythmias in the context of HF should be mentioned

Author Response

Reviewer-1

The Authors of this review focuses on the HF pharmacotherapy in children mainly using extrapolations from adult cardiology practices or guideline suggestions. The review is well written. However, to increase its clinical impact the Authors should address the following issues:

1) The therapy should be addressed according to HF categorization, including reduced or preserved ejection fraction phenotypes. Some specific heart diseases subtending HF in children may deserve brief comment (e.g., obstructive hypertrophic cardiomyopathy)

- Thank you for your comments. We have described our approach to Acute and Chronic HF. We included only HF due to reduced ejection fraction (HFrEF) in this review. We clarified this point in the introduction section.

This review is a follow-up of the previous review article on Update in HF: published in Children in 2018. (June 2018 Children 5(7):88 ; DOI: 10.3390/children5070088. In the previous review, different etiologies of HF have been discussed, and readers can refers to the cited paper to gain knowledge about different types of HF. We primarily focused on the drugs and devices available for HFrEF and future goals to pursue clinical trials in children using adult clinical trial models.

- We try to avoid specific etiologies for HF in children as it is heterogeneous. Therefore, we adopted a general approach to acute and chronic HF. The reviewer suggested discussing management for HF due to obstructive hypertrophic cardiomyopathy, and it is a distinct phenotype and is beyond the scope of this review paper.

2) Pharmacodynamics peculiarities and recommended doses of HF medications in children vs. adults deserve comment

- Thank you for your comments. We have added in the introduction that pharmacokinetics and pharmacodynamics characteristics of drugs in adults are different from children. Pediatric studies are needed to navigate these drugs and devices available for HF in adults.

-We added doses when relevant e.g., under valsartan and sacubitril: “tablets are available in three dosage strengths: 24/26 mg, 49/51 mg, and 97/103 mg (sacubitril/valsartan). These doses are referred to as 50 mg, 100 mg, and 200 mg in the clinical trial literature, including The New England Journal of Medicine publication of PARADIGM-HF results (Ref #47). An oral solution also may be compounded. In adult patients, the target maintenance dose of Entresto is 97/103 mg twice daily as tolerated by the patient. In pediatric patients, the target maintenance dose is dependent on body weight.” Table-2 is added for doses of commonly used pediatric drug doses. 

3) The treatment of arrhythmias in the context of HF should be mentioned.

- Thank you for your comments. This is an important point, and we have discussed how arrhythmia can affect the HF and discussed under β-blockers, digoxin, and Ivabardine headings, how to control heart rate. We also added under device therapy to treat autonomic dysfunction.

Reviewer 2 Report

This is a good article. However, it does not address the core issue of "pediatric" medicine: what is a child? Persons <18 years of age are legally and administratively still children, but not all of them bodily. This is highly relevant in this context. Chapter 3.4. on the angiotensin receptor - neprilysin Inhibitor describes the PARADIGM-HF study with 3 strata: 1 month to 1y, 1y to 6y, 6y to 17y. In this study, the adolescents were bodily already mature. Therefore, the basis of this study was flawed. The manuscript describes that FDA approved sacubitril/valsartan for use in children older than 1 year of age with symptomatic HF with reduced systolic function, a decision which medically makes sense. The only flaw in the FDA's decision is that it still classifies individuals 17 and under as "children". But medicine and drugs  deal with physical persons, not with their legal or administrative status. 

The authors should critically discuss the inclusion criteria of the PARADIGM-HF study and should discuss the FDA decision in this context.

The authors might consider to read "Considering the Patient in Pediatric Drug Development", https://www.elsevier.com/books/considering-the-patient-in-pediatric-drug-development/rose/978-0-12-823888-2

Author Response

This is a good article. However, it does not address the core issue of "pediatric" medicine: what is a child? Persons <18 years of age are legally and administratively still children, but not all of the bodily. This is highly relevant in this context. Chapter 3.4. on the angiotensin receptor - neprilysin Inhibitor describes the PARADIGM-HF study with 3 strata: 1 month to 1y, 1y to 6y, 6y to 17y. In this study, the adolescents were bodily already mature. Therefore, the basis of this study was flawed. The manuscript describes that FDA approved sacubitril/valsartan for use in children older than 1 year of age with symptomatic HF with reduced systolic function, a decision which medically makes sense. The only flaw in the FDA's decision is that it still classifies individuals 17 and under as "children". But medicine and drugs deal with physical persons, not with their legal or administrative status. 

The authors should critically discuss the inclusion criteria of the PARADIGM-HF study and should discuss the FDA decision in this context.

  • Thank you for your comments. We think the reviewer refers to the PANORAMA-HF trial. We critically discussed on FDA's decision to approve valsartan/Sacubitril in children >1 year based on the findings through week 12, from the 52-week trial in which cardiac biomarker NT-proBNP was reduced among pediatric patients. Because sacubitril/valsartan was shown to improve cardiovascular outcomes and reduce NT-proBNP among adult patients in the PARDAIGN-HF trial, the observed effect on the drug on NT-proBNP was deemed reasonable based to improve similar outcomes in pediatric patients. Moreover, the safety and tolerability in pediatric patients were consistent with that observed in adult patients.

The authors might consider reading "Considering the Patient in Pediatric Drug Development", https://www.elsevier.com/books/considering-the-patient-in-pediatric-drug-development/rose/978-0-12-823888-2

  • Thank you for this useful reference. As the reviewer pointed out: A child is not an adult. The drugs used in adults can harm children as the maturation of the human body regarding absorption, distribution, metabolism, and excretion results in key differences between newborns, infants, older children, and adolescents. We completely agree with the reviewer that there is a need to do drug trials in children. This review paper is exactly discussing the same issue. We want to emphasize following the PANORAMA trial: it is feasible to conduct drug trials in children with appropriate end-points and the lessons that we can learn from other adult trials to pursue meaningful trials in children in the future.

Round 2

Reviewer 2 Report

Authors should have discussed how the define children. The American Academy of Pediatrics defines "children" as up to 21 years; for the Food and Drug Administration (FDA) and the European Medicines Agency (EMA) "children" are <18 years old. But the 18th birthday is an administative, not a physiological age limit. The body matures slowly during puberty, which usually is over well ahead of the 18th birthday. Thus, the authors' research approach and conclusions are caught within a flawed framework.

Author Response

The focus of this review is to discuss the current and future treatment of HF in children and lessons learned from adult trials. 

How to define children is a philosophical issue. We are not going to confuse readers on this controversial topic. We request readers to read an excellent book by our Academic editor on this topic, 

"Considering the Patient in Pediatric Drug Development", http://www.elsevier.com/books/considering-the- patient-in-pediatric-drug-development/rose/978-0-12-823888-2.